# Chiral two-dimensional conjugated metal-organic frameworks with high spin polarization

Shiyi Feng[1,11], Yang Lu[1,2,11] ✉, Chenchen Wang[1,3], Morteza Torabi[1,4], Xing Huang[1], Florian Auras[1], Ran He[5], Lukas Sporrer[1,6], Paul-Alexander Laval-Schmidt[1], Xizheng Wu[7], Xia Wang[7], Li Wan[6], Dongxu Wang[6], Bernd Plietker[8], Mike Hambsch[9], Markus Löffler[10], Stefan C. B. Mannsfeld[9], Claudia Felser[7] & Xinliang Feng[1,6] ✉

Two-dimensional conjugated metal-organic frameworks have garnered significant research interest as emerging candidates for organic 2D crystal materials. In classical 2D crystals, chirality can give rise to unique physical phenomena; however, the precise implantation of chirality into 2D c-MOFs has yet to be demonstrated. Here, we demonstrate an example of chiral 2D c-MOFs obtained through the side chain-induced chirality amplification strategy to achieve tunable chiral expression, enabling notable chirality amplification. Chiral substitutions induce distortions in the 2,3,7,8,12,13-hexaiminotriindole conjugated ligand along distinct orientations, effectively transferring their chirality to the conjugated backbone. The resulting frameworks exhibit tunable chirality through steric control, where chiral naphthylethyl substitution increases the distortion angle from 2.3° to 7.0°, leading to pronounced chirality amplification. Furthermore, the chiral 2D c-MOFs demonstrate exceptional spin polarization measured by magnetic-conductive atomic force microscopy (mc-AFM), achieving a value as high as 96.9%, placing them among the highest-performing materials reported to date. This work opens new avenues for the design of chiral 2D crystal materials with potential applications in chiral spintronics.

Chiral two-dimensional (2D) crystals have garnered significant interest over the past decades due to their potential in diverse applications, including opto-electronics[1,2] and spintronics[3,4], and chirality-induced phenomena[5,6]. In particular, chirality-induced spin selectivity (CISS) effect has been widely investigated as an intriguing physical phenomenon, which refers to the ability of chiral molecules to selectively transmit electrons with a specific spin orientation, resulting in substantial spin polarization[7]. This effect has been extensively studied in

[1]Center for Advancing Electronics Dresden, Faculty of Chemistry and Food Chemistry, Technische Universität Dresden, Dresden, Germany. [2]Key Laboratory of Low-grade Energy Utilization Technologies and Systems, National Innovation Center for Industry-Education Integration of Energy Storage Technology, School of Energy and Power Engineering, Chongqing University, Ministry of Education, Chongqing, China. [3]Institute for Materials Science, Max Bergmann Center for Biomaterials, Technische Universität Dresden, Dresden, Germany. [4]Department of Organic Chemistry, Faculty of Chemistry and Petroleum Sciences, Bu-Ali Sina University, Hamedan, Iran. [5]Leibnitz Institute for Solid State and Materials Research Dresden e.V., (IFW-Dresden), Dresden, Germany. [6]Max Planck Institute of Microstructure Physics, Halle, Germany. [7]Max Planck Institute for Chemical Physics of Solids, Dresden, Germany. [8]Chair of Organic Chemistry I, Faculty of Chemistry and Food Chemistry, Technische Universität Dresden, Dresden, Germany. [9]Center for Advancing Electronics Dresden, Faculty of Electrical and Computer Engineering, Technische Universität Dresden, Dresden, Germany. [10]Dresden Center for Nanoanalysis (DCN), Center for Advancing Electronics Dresden (cfaed), Technische Universität Dresden, Dresden, Germany. [11]These authors contributed equally: Shiyi Feng, Yang Lu. ✉e-mail: yang.lu@cqu.edu.cn; xinliang.feng@mpi-halle.mpg.de

various materials, including chiral organic molecules[8–10], chiral inorganic nanostructures[11–13], chiral polymers[14–16], and chiral covalent organic frameworks (COFs)[17–19]. Chiral systems with high spin polarization are especially promising for spintronic filters, where spin-dependent electron transport is key to enhancing the selectivity[20]. However, despite the critical role of structural chirality, most reported chiral materials display spin polarization below 90%, largely limited by their poor electrical conductivity[10,14,21–25].

In recent decades, 2D conjugated metal-organic frameworks (2D c-MOFs) have emerged as a new class of organic 2D crystal materials[26–28]. These materials combine inherent in-plane electronic conjugation and out-of-plane coupling together with exceptional chemical tunability, diverse topological structures, and intrinsic spin-related properties[27,29–31]. Particularly, the unique d-π conjugated networks formed by π-conjugated organic linkers and metal ions induce strong spin-orbit coupling, thereby enhancing intrinsic spin polarization[26,32,33]. Thus, implanting chirality into 2D c-MOFs represents a promising strategy for constructing spin filter materials with high spin-selectivity and polarization efficiency.

Efforts to introduce chirality into inherently achiral 2D materials have primarily focused on chiral molecules intercalation, wherein chiral species are intercalated between adjacent layers of van der Waals 2D materials to impart chirality to the bulk structure[34,35]. However, this strategy often suffers from issues such as layer-to-layer inhomogeneity and interlayer interference, largely limiting the control over chirality modulation and lacking robustness of the resultant chiral 2D materials[36–38]. Alternatively, other studies have attempted to induce chirality into framework materials by incorporating asymmetric building blocks, as demonstrated in chiral COFs[39–41] and chiral metallacycles[42]. Nevertheless, this approach inherently conflicts with the design principles of 2D c-MOFs, which predominantly rely on high-symmetry ligands to construct extended 2D networks[43]. Consequently, implanting chirality into 2D c-MOFs remains a significant challenge, largely due to the lack of chiral building blocks compatible with the symmetrical architecture essential for 2D c-MOF construction.

Herein, we present a side chain-induced chirality amplification (SICA) strategy for synthesizing conjugated ligands with intrinsic molecular chirality. This approach allows precise control over chiral expression and facilitates chirality amplification in 2D c-MOFs, while maintaining high framework symmetry. By grafting chiral molecules onto the nitrogen atoms of π-conjugated 2,3,7,8,12,13-hexaimino-triindole (HATI) ligands, the introduced chiral substitutions induce a distortion in the 2D planes of the aromatic backbones, leading to fan-like shapes with varying orientations that exhibit a molecular propeller-like feature, as shown in Fig. 1a. Consequently, the chirality of the side chains can be transferred to the conjugated backbone. Upon constructing the secondary building units (SBUs) with Ni ions, the synthesized 2D c-MOFs exhibit high crystallinity and pronounced chirality, suggesting the retention of chirality from ligand molecules to the uniform spatial organization in the MOF crystals. Notably, the degree of chirality can be tuned by adjusting the steric hindrance of the chiral monomers. For instance, replacing chiral phenylethyl (PhEt) with chiral naphthylethyl (NaEt) increases the distortion angle from 2.3° to 7.0°, leading to enhanced chirality amplification of the bulk material. This structural chiral amplification directly contributes to improved spin polarization in the chiral 2D c-MOF systems. As a proof-of-concept, we employ magnetic-conductive atomic force microscopy (mc-AFM) to evaluate the spin polarization properties of these chiral 2D c-MOFs. These samples feature intrinsic molecular chirality and exceptional electrical conductivity, achieving a polarization value as high as 96.9%-among the highest reported for chiral materials. Our SICA strategy offers a robust pathway for designing chiral 2D crystal materials with high conductivity and spin polarization performance, highlighting the significant potential of chiral 2D c-MOFs in chiral electronics and spintronics.

## Results

The HATI-conjugated ligand offers a straightforward approach for the introduction of chiral groups onto the modifiable nitrogen atoms of the indole unit. Chiral PhEt and chiral NaEt groups with different configurations are introduced into HATI ligands via the Mitsunobu reaction, yielding HATI-(S)-PhEt, HATI-(R)-PhEt, and HATI-(S)-NaEt, respectively (where S or R corresponds to different chirality, as detailed in the Supplementary Information) (Fig. 1b)[44,45]. The introduction of chiral PhEt and NaEt on the ligands brings the distortion of the molecular structure, resulting in a characteristic molecular propeller shape as predicted by density functional theory (DFT) calculations (Fig. 1b). Chiral groups with different configurations induce the ligand backbone to distort into a fan-like shape, with rotational direction corresponding to their chirality. It is notable that the chiral group with larger steric hindrance enhances molecular distortion. According to the DFT calculations, substituting the chiral PhEt group with the chiral NaEt group raises the distortion angle from 2.3° to 7.0°, leading to the formation of larger fan blades. Coordination polymerization between chiral HATI ligands and $Ni^{2+}$ ions in mixtures of dimethyl sulfoxide (DMSO) and water at 65 °C for 2 hours yielded chiral 2D c-MOFs, designated as (S)-$Ni_3$(HATI_PhEt)$_2$, (R)-$Ni_3$(HATI_PhEt)$_2$, and (S)-$Ni_3$(HATI_NaEt)$_2$, simplified as (S)-Ph, (R)-Ph, and (S)-Na, respectively. Then, the chirality of ligands is expected to be transferred and aggregated, resulting in the formation of chiral 2D c-MOFs (Fig. 2a).

The crystalline structures of chiral 2D c-MOFs were determined by powder X-ray diffraction (PXRD) analysis with Cu Kα radiation. Figure 2b shows the PXRD patterns of chiral 2D c-MOFs under their optimal synthesis conditions. The major diffraction peaks observed in the low-angle region (3.7°, 6.5°, 7.6°, 10.0°, 13.2°) appear at the same position for all chiral 2D c-MOFs, corresponding to the (100), (110), (020), (210) and (220) reflection planes, respectively. This result suggests that the in-plane crystalline ordering remains consistent with different chiral side groups. Due to the varying steric hindrance of chiral substituents, the diffraction peaks at around 28.0°, 28.0°, and 27.5° for (S)-Ph, (R)-Ph, and (S)-Na, respectively, correspond to interlayer π-π spacings of 3.26, 3.26, and 3.36 Å, respectively. DFT calculations were performed to elucidate the structures of chiral 2D c-MOFs and determine the unit cell parameters. The experimental XRD patterns were well reproduced using an AA-eclipsed stacking geometry. Specifically, the aromatic rings on the side chains contribute to interlayer π-π stacking, which stabilizes the twisted conformation and helps maintain both the chiral framework and its rigidity. Pawley refinements yielded optimized parameters with good consistency factors, as shown in Supplementary Fig. 1 ($R$p = 2.70% and $R$wp = 1.80% for (S)-Ph; $R$p = 4.01% and $R$wp = 2.33% for (R)-Ph; $R$p = 2.74% and $R$wp = 1.62% for (S)-Na). The experimental PXRD patterns exhibited similar peak widths at half maximum, suggesting comparable crystallinity among the three samples. Scanning electron microscopy (SEM) revealed that the chiral 2D c-MOF powders consist of rod-like crystals (Fig. 2c and Supplementary Fig. 2). Structures of chiral 2D c-MOFs were further demonstrated by high-resolution transmission electron microscopy (HRTEM). Clear lattice fringes and a well-defined honeycomb lattice were observed from out-of-plane and in-plane directions (Fig. 2d, e and Supplementary Fig. 3). Fast Fourier transform (FFT) analysis determined lattice distances of 24.3 Å, 24.3 Å, and 25.2 Å for (S)-Ph, (R)-Ph, and (S)-Na, respectively, which are consistent with the PXRD and simulated results.

The porosities of chiral 2D c-MOFs were investigated by $N_2$ adsorption isotherms at 77 K. (S)-Ph, (R)-Ph, and (S)-Na showed the BET surface area of as high as 1440 m$^2$ g$^{-1}$, 1215 m$^2$ g$^{-1}$, and 999 m$^2$ g$^{-1}$, respectively (Supplementary Fig. 4). The pore size of ~1.86 nm is shared by these three samples and fitted by the QSDFT model. High-resolution analysis of the Ni 2$p$ X-ray photoelectron spectra (XPS) confirmed the presence of Ni (II) in all chiral 2D c-MOFs (Supplementary Fig. 5). Fourier-transform infrared (FT-IR) spectroscopy revealed the disappearance of the N-H stretching vibration band of the chiral ligands

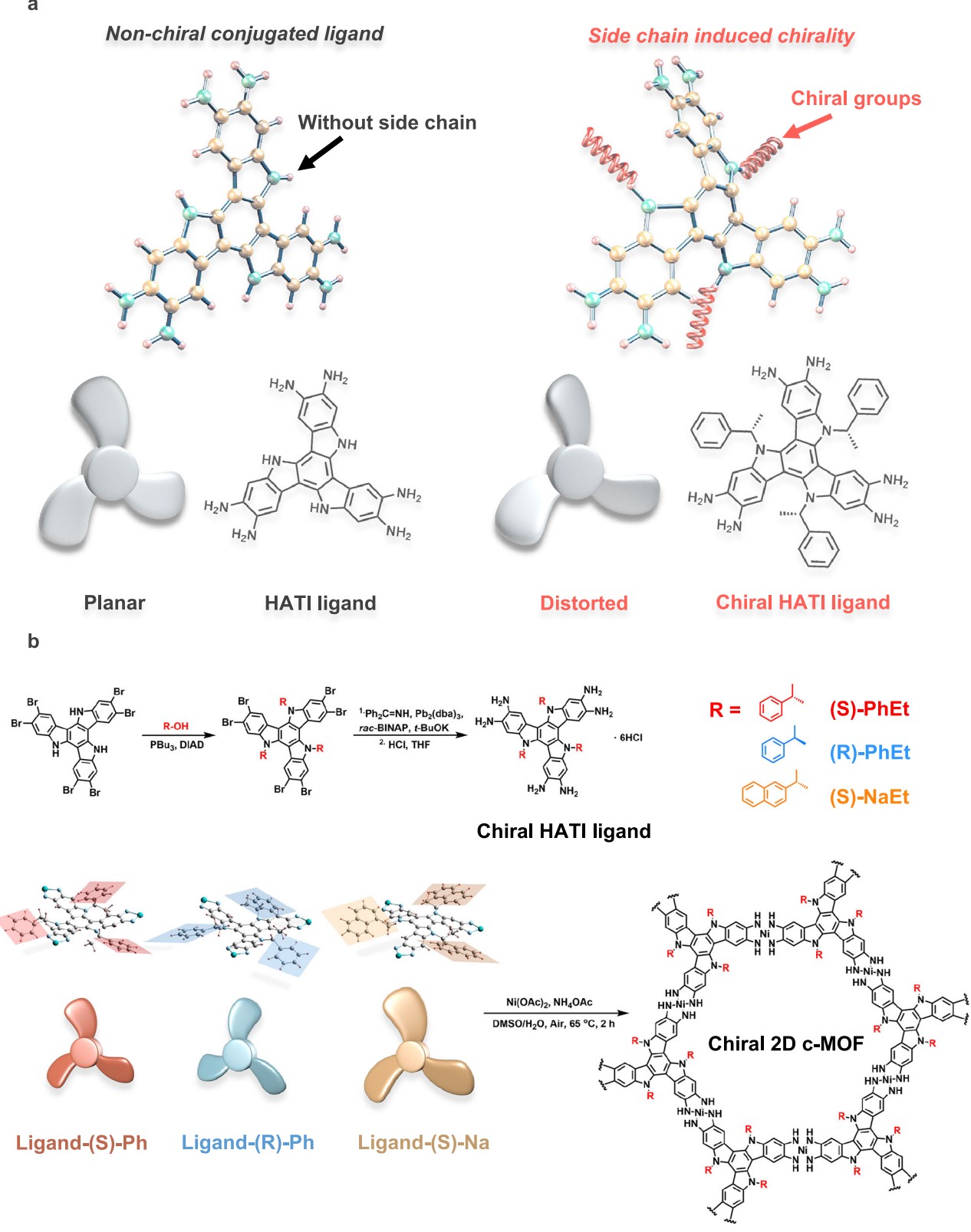

**Fig. 1 | Design, synthesis, and structure elucidation of chiral ligands.**
**a** Schematic illustration of the strategy of side chain induced chirality and the structure of planar HATI ligand and distorted chiral HATI ligand. **b** Synthesis of chiral HATI ligands and corresponding chiral 2D c-MOFs with different chiral substitutions. Ligand-(S)-Ph denoted by red color, ligand-(R)-Ph denoted by blue color, and ligand-(S)-Na denoted by yellow color.

after MOF synthesis, confirming efficient coordination polymerization (Supplementary Fig. 6). Thermogravimetric analysis (TGA) indicated that all chiral 2D c-MOFs exhibited significant weight loss due to decomposition beyond 200 °C (Supplementary Fig. 7).

As previously discussed, the incorporation of chiral PhEt and NaEt groups into the HATI ligands induces molecular distortion, which is responsible for the chiral characteristics. As illustrated in Supplementary Fig. 8 and Fig. 3a, the circular dichroism (CD) spectra of the

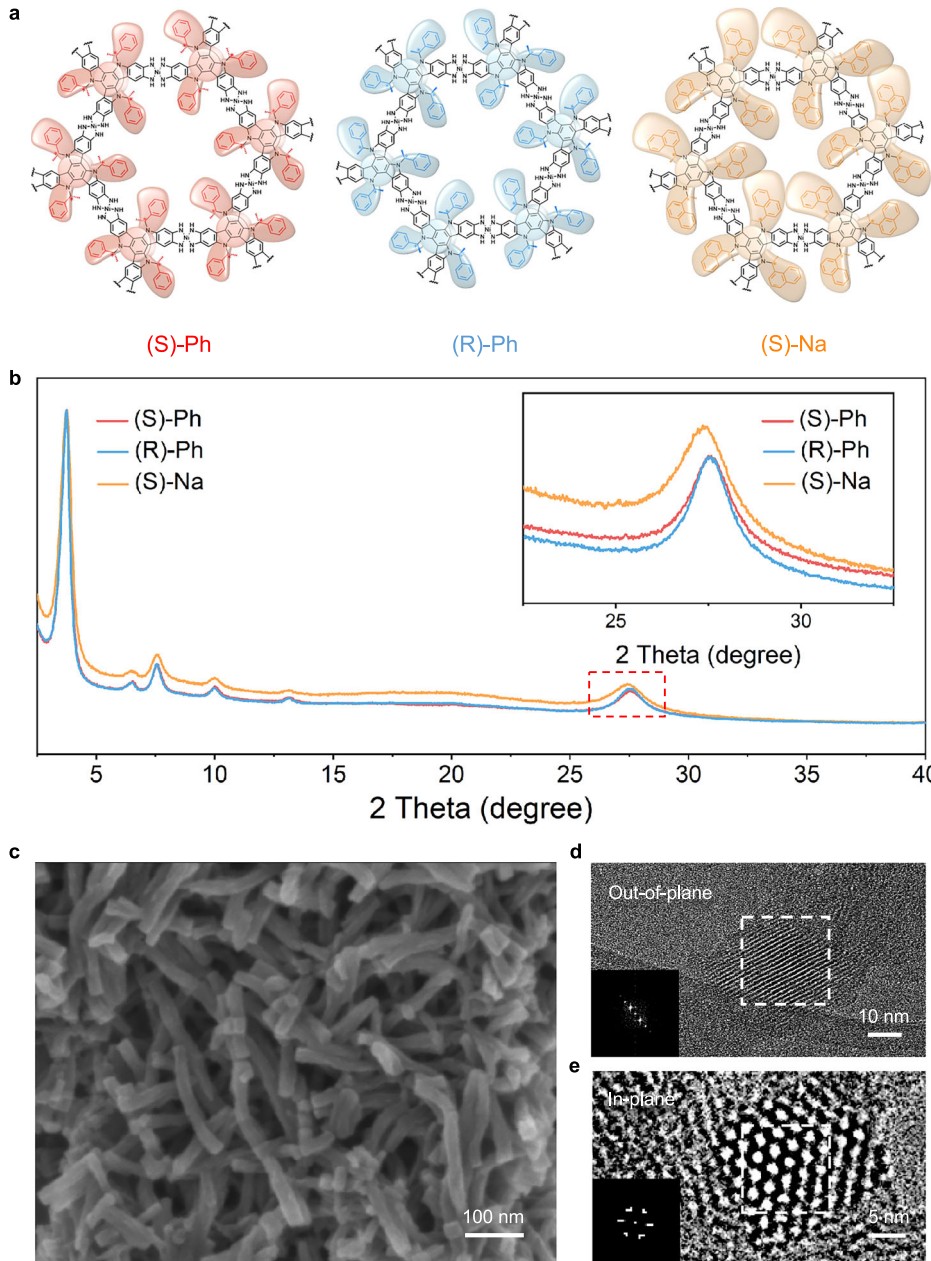

**Fig. 2 | Structure elucidation of chiral 2D c-MOFs. a** Schematic diagram of chiral 2D c-MOF structure. **b** PXRD pattern of chiral 2D c-MOFs (inset: an enlarged view of the (00 l) diffraction peaks). **c** SEM image of (R)-Ph. **d** HRTEM image of (R)-Ph from side-view (inset: FFT analysis). **e** HRTEM image of (R)-Ph from top-view (inset: FFT analysis).

chiral PhEt groups and the chiral ligands exhibit perfect mirror behavior, confirming the successful transfer of chirality from the small molecules to the ligands. The observed Cotton effects in the 250–300 nm range originate from the chiral PhEt groups, while four additional intense bands appearing in the 300–400 nm region are attributed to the core HATI units. Upon coordination of the chiral HATI ligands with Ni ions, the resulting 2D c-MOF powder dispersions exhibit strong Cotton effects with clear CD signals (Fig. 3b). Notably, the two major bands of the chiral 2D c-MOFs at 280 and 370 nm are red-shifted relative to the corresponding bands at 273 and 323 nm in the chiral ligands, suggesting the successful propagation of chirality from the ligands to the MOF backbone. Moreover, the ligands with (S)-NaEt on the side chain exhibit stronger Cotton effects compared to (S)-PhEt, which can be attributed to the increased steric hindrance by the increased structural distortion (Fig. 3c, d). These intense bands align well with the corresponding UV-vis spectra (Supplementary Fig. 9). Notably, these chiral 2D c-MOFs exhibit excellent ambient and

moisture stability, retaining high crystallinity after 30 days of storage at room temperature (Supplementary Fig. 10). In contrast, while the racemic samples exhibit comparable absorption to their chiral counterparts, they show no CD signals and Cotton effects.

Notably, despite the presence of different chiral groups, the electrical conductivities of the chiral 2D c-MOFs exhibited minimal variation. Parallel four-probe measurements conducted under ambient conditions revealed electrical conductivities of 5.43 S m⁻¹, 5.27 S m⁻¹, and 3.35 S m⁻¹ at 298 K for (S)-Ph, (R)-Ph, and (S)-Na powder pellets, respectively (Supplementary Fig. 11). Variable-temperature conductivity measurements performed between 200 and 300 K indicated thermally activated charge transport for all three samples. The hopping activation energies were determined to be 93.9, 99.1, and 103.3 meV for (S)-Ph, (R)-Ph, and (S)-Na, respectively, suggesting comparable charge transport behavior among these three 2D c-MOFs.

To experimentally explore their CISS, we employed mc-AFM to examine the spin polarization behavior of the materials under ambient

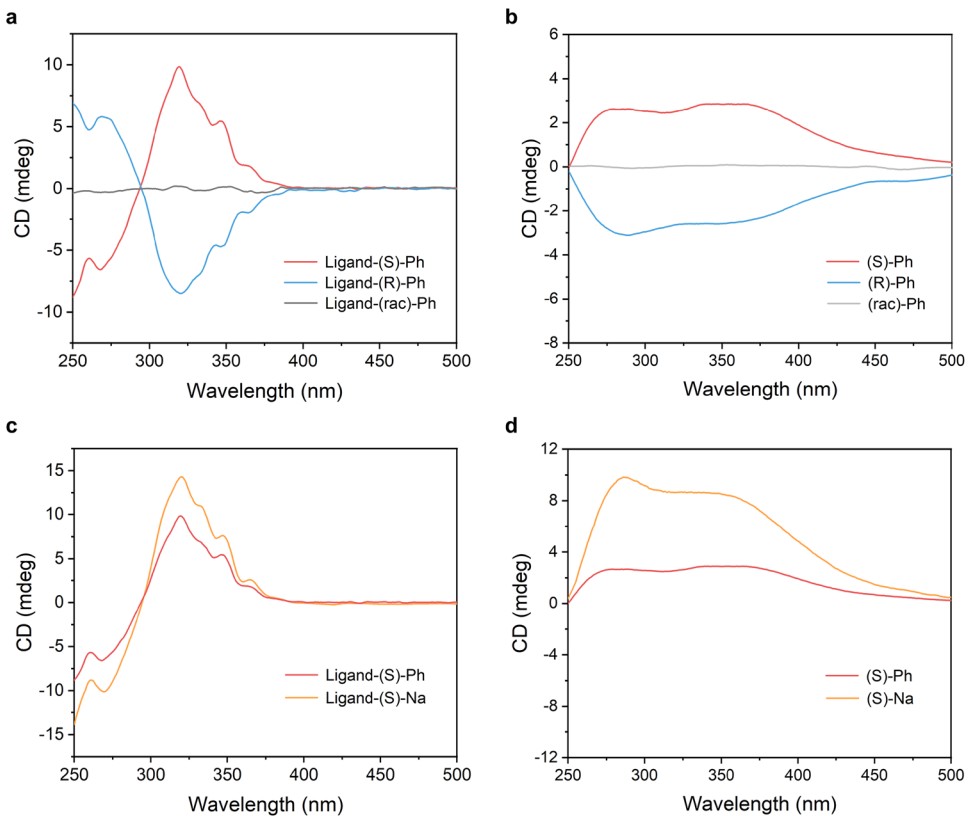

**Fig. 3 | CD spectra of chiral ligands and the resultant chiral 2D c-MOFs. a, c** CD spectra of chiral ligands with (S)-PhEt, (R)-PhEt, (S)-NaEt and (rac)-PhEt substitutions. **b, d** CD spectra of chiral 2D c-MOFs, (S)-Ph, (R)-Ph, (S)-Na and (rac)-Ph.

conditions. However, the high contact resistance between the substrate and nanocrystals hindered accurate measurements. To address this issue as a proof-of-concept, chiral 2D c-MOF films were synthesized with thicknesses of 101.3, 100.8, and 101.7 nm for (S)-Ph, (R)-Ph, and (S)-Na, respectively, using Ni/Au substrates for subsequent CISS measurements (see Supplementary Information, Supplementary Figs. 12–19 for synthesis details). The pronounced chirality of these chiral 2D c-MOF films offers a significant potential for enhanced spin polarization and selectivity. The spin polarization ratio, defined by the difference in current intensity through a chiral medium when the magnetization direction is reversed, offers a direct measure of spin selectivity. The use of highly crystalline chiral MOF films in CISS devices significantly reduces contact resistance, thereby improving both spin polarization and selectivity (Fig. 4a). The excellent electrical conductivity of chiral MOF films greatly facilitates efficient current flow, enabling effective interactions with chiral systems. The current measured in the I-V experiments using mc-AFM reached several microamperes, surpassing values previously reported for chiral systems and significantly higher than values previously reported for chiral systems in CISS research (Fig. 4b–d). With upward magnetization, the current in (S)-Ph increased sharply as the bias voltage rose. Conversely, under downward magnetization, the current remained relatively unchanged, indicating that (S)-Ph exhibited higher resistance with downward magnetization. In contrast, the current response in (R)-Ph was opposite to that of (S)-Ph under the same conditions. The degree of spin polarization was further quantified using the following equation:

$$P = \left[ \left( \frac{I_{up} - I_{down}}{I_{up} + I_{down}} \right) \right] \times 100\% \qquad (1)$$

where $I_{up}$ and $I_{down}$ represent the current detected by mc-AFM with the substrate magnetized upward and downward, respectively[19]. Due to the excellent electrical conductivity and superior spin selectivity, the spin polarization values for (S)-Ph and (R)-Ph were measured at 91.6% ± 0.3% and −91.1% ± 0.5%, respectively. Notably, substituting the chiral group with one or more significant steric hindrances led to an even higher spin polarization for (S)-Na, reaching 96.9% ± 0.3%. This represents one of the highest values reported for spin polarization in existing materials. Significantly, the spin current in the (S)-Na film reached ~3000 nA, exceeding previously reported values for CISS materials (Supplementary Fig. 20 and Supplementary Table 1)[46]. These remarkable spin current values align well with the measured electrical conductivities, where the chiral substitutions have minimal impact on the bulk electrical conductivity and charge transport properties of the 2D c-MOF materials. This pronounced chiral amplification effect likely originates from the structural dissymmetry introduced by the steric hindrance of the chiral groups. Additionally, we conducted over 100 independent measurements for each sample batch, consistently observing stable and reproducible current values. This result suggests that the high electrical conductivity of chiral 2D c-MOFs is crucial for ensuring reliable spin polarization results, particularly in terms of data reproducibility. We propose that the differential tunneling conductance between the two enantiomers is influenced by the spin selectivity of the substrate and the chirality of the MOF. Specifically, an upward-magnetized substrate facilitates efficient transport of spin-polarized charge carriers between the substrate and the film. In contrast, a downward-magnetized substrate disrupts this flow, leading to increased resistance. Thus, increasing the steric bulkiness from chiral PhEt to chiral NaEt groups induces more significant distortion in the 2D planes, which enhances spin selectivity, ultimately resulting in a higher spin polarization value.

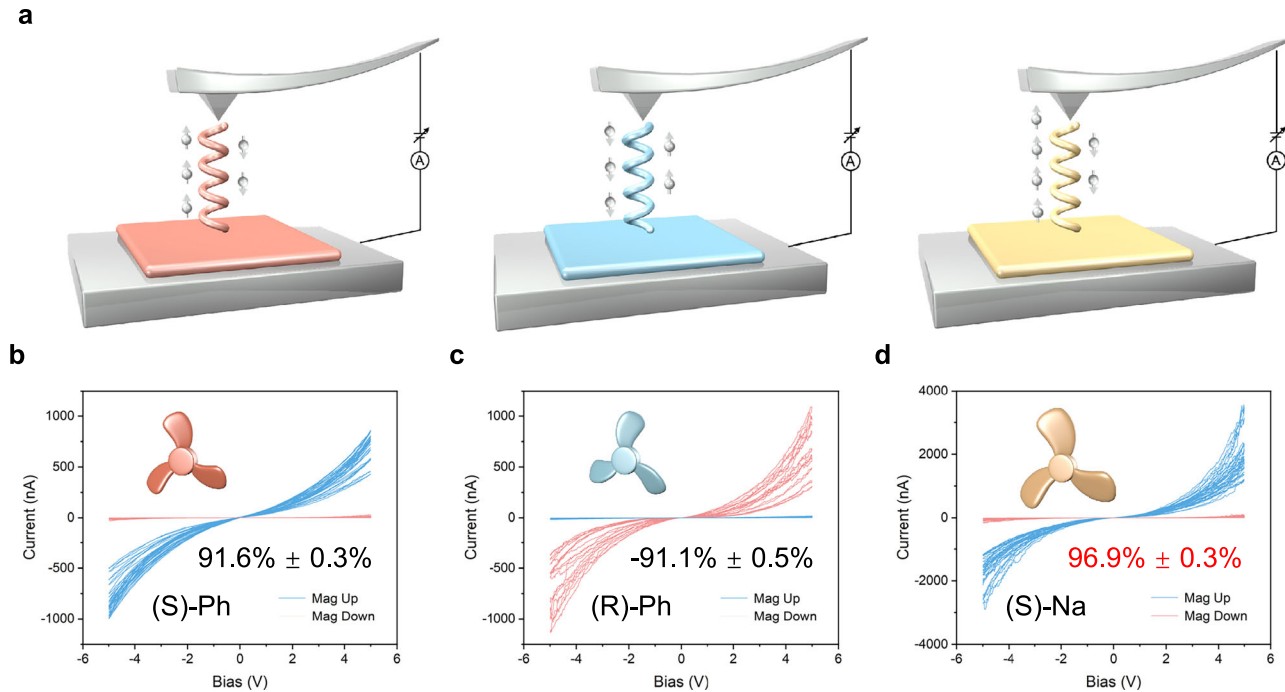

**Fig. 4 | Spin polarization of chiral 2D c-MOFs with prominent spin filter properties. a** Schematic illustration of the tunneling current in a parallel configuration for (S)-Ph, (R)-Ph, and (S)-Na. **b–d** Room-temperature I–V curves for chiral 2D c-MOF films at thickness -100 nm obtained by mc-AFM.

Moreover, we conducted thickness-dependent CISS measurements using films with thicknesses of 31.5 nm, 101.3 nm, and 308.5 nm (Supplementary Fig. 14). As shown in Supplementary Fig. 21, the measured spin polarization values of (S)-Ph at thicknesses of 31.5 nm, 101.3 nm, and 308.4 nm are 86.8% ± 0.2%, 91.6% ± 0.1%, and 95.6% ± 0.1%, respectively. The spin polarization in the chiral MOF films exhibits a clear dependence on thickness, which correlates with the length of the chiral structure. These results, including both the sign of the polarization and its dependence on length, are in good agreement with previously reported studies[8,10,47,48]. This strong thickness dependence is likely due to the extended length of the chiral structures within the MOF films. The inherent face-on stacking arrangement enables charge carriers to move along the out-of-plane direction, promoting efficient transport of spin-polarized current through the continuous chiral pathways and ensuring effective interaction with the chiral environment[49,50]. In addition, the high crystallinity of the chiral MOF films facilitates long-range charge transport, which contributes to the observed ultrahigh spin polarization values.

## Discussion

In summary, we develop a SICA strategy as an effective approach to introduce and amplify chirality in 2D c-MOFs. By functionalizing π-conjugated HATI ligands with chiral substituents of varying steric bulkiness and coordinating them with metal ions, we achieve effective chirality transfer from the molecular level to the extended 2D frameworks. Increasing the steric hindrance from chiral PhEt to chiral NaEt induces a notable increase in distortion angle from 2.3° to 7.0°, leading to enhanced chirality amplification in the resulting materials. As a result, we developed the chiral 2D c-MOFs exhibiting excellent electrical conductivity and a remarkable spin polarization of 96.9%, among the highest reported for chiral materials. The synthesis of high-crystallinity chiral 2D c-MOF films, along with the fabrication of high-performance spintronic devices, will remain the focus of our future efforts. This design strategy lays a solid foundation for the development of chiral organic 2D crystal materials and opens promising avenues for future applications in chiral electronics and spintronics.

## Methods

### Nuclear magnetic resonance (NMR) spectroscopy
NMR spectra were recorded on Bruker AV-II 300 (300.1 MHz for 1 H). Chemical shifts are reported in ppm relative to TMS.

### FT-IR spectroscopy
FT-IR spectra were acquired using a Bruker Optics ALPHA-E spectrometer with a Zn-Se ATR accessory, covering the 400–4000 $cm^{-1}$ range.

### Ultraviolet-visible (UV-Vis) spectroscopy
UV-Vis absorption measurements were performed at room temperature using an Agilent Cary 5000 UV-Vis-NIR spectrophotometer.

### Powder X-ray diffraction (PXRD)
PXRD patterns were collected on an Aeris Research Edition diffractometer (Malvern Panalytical) with Cu-Kα radiation ($\lambda = 0.15418$ nm) at 40 kV and 15 mA in reflection mode at room temperature.

### Scanning electron microscopy
SEM images were obtained using a Zeiss Gemini S4500 microscope.

### Transmission electron microscopy
TEM images were captured on a JEOL JEM F200 microscope.

### Optical microscopy
Optical images were captured on a Zeiss optical microscope.

### Thermogravimetric analysis
TGA measurements were conducted on a TA Instruments Q600 analyzer under a nitrogen atmosphere with a heating rate of 5 °C/min using ceramic crucibles. Prior to analysis, powder samples were dried in a supercritical $CO_2$ dryer for 4 hours and subsequently activated at 90 °C overnight. During the TGA measurement, in-situ activation at 100 °C for 1 hour was performed.

## Circular dichroism

CD spectra were measured on Chiralscan V100 spectrophotometers. The chiral molecules and chiral ligand samples were prepared by dissolving them in chloroform. The chiral MOF powders were prepared by dispersing them in ethanol to form a homogenous dispersion. The chiral MOF films were first transferred onto a glass slice and then fixed in the sample tube. The data were collected with a scanning rate of 500 nm/min, ranging from 240 to 500 nm, at a temperature of 293 K.

## Gas adsorption analysis

Nitrogen sorption isotherms were recorded at 77 K on a Quantachrome BELSORP adsorption analyzer. Surface areas were calculated based on the Brunauer-Emmett-Teller (BET) theory. Pore size distributions were derived from the adsorption isotherms using the quenched solid density functional theory (QSDFT) equilibrium model. Prior to analysis, samples were dried in a supercritical $CO_2$ dryer for 4 hours and then activated at 90 °C overnight.

## X-ray photoelectron spectroscopy

XPS measurements were carried out on a Thermo Scientific K-Alpha Spectrometer under an ultrahigh vacuum ($-3 \times 10^{-9}$ Torr) using a monochromatic Al Kα source (1486.6 eV, θ = 90°, operated at 14 kV and 15 mA). Instrumental energy resolution was 0.5 eV for XPS.

## Grazing-incidence wide-angle X-ray scattering (GIWAXS)

The GIWAXS measurements of chiral MOF films were conducted at beamline 11-NCD-SWEET at ALBA (Spain). The incident X-ray beam had an energy of 12.4 keV and a beam size of 20 μm (vertical) × 70 μm (horizontal). Data were collected using a Rayonix LX255HS detector, positioned 181 mm behind the sample. The sample-to-detector distance and the beam center position on the detector were calibrated using a chromium (III) oxide standard. Measurements were performed at an incident angle of 0.12°, and five images with 10-second exposure times were recorded to enable averaging. Data correction and analysis were carried out using the WxDiff software.

## Electrical conductivity

Pressed pellets were prepared by applying 10 mg of sample onto a polymer film in a split-sleeve press at room temperature. Pellets were then heated at 150 °C under vacuum for 2 hours for complete desolvation. After desolvation, the pellet thickness was measured, and four silver wire probes were attached to the top using conductive silver paste. The device was kept in the air for 1 hour to allow the paste to dry completely. Electrical conductivity was measured using the van der Pauw geometry on a Lakeshore Hall System (9700 A). Activation energies were calculated using the Arrhenius equation: $\sigma (T) = \sigma_0 \exp(-E_a/k_B T)$, where $\sigma_0$ is the pre-factor and $k_B$ is the Boltzmann's constant.

## Magnetic-conducting atomic force microscope

**Sample preparation.** A silicon wafer with a 500 nm thermal oxide layer was used as the substrate. A 5 nm titanium adhesion layer and an 80 nm nickel layer were sequentially deposited, followed by a 5 nm gold protective layer. The resulting Ni/Au substrate facilitates spin polarization of electrons injected into the chiral MOF films under an external magnetic field. Prior to film transfer, the substrate was sequentially cleaned by immersion in ethanol and acetone for 5 minutes each, then dried under a nitrogen stream. Chiral 2D c-MOF films were subsequently transferred onto the prepared Ni/Au substrate for mc-AFM measurements.

## Measurement of the CISS effect

Current-voltage (I–V) characteristics were measured using a conductive atomic force microscopy (c-AFM) mode on a Park NX10 system (Park Systems). Measurements were performed in pin-point mode, in which a voltage bias is applied through the conductive tip while minimizing lateral forces, thereby enhancing spatial resolution and reducing sample deformation or displacement. A platinum-coated conductive probe (model: 25Pt300B; resonance frequency: 20 kHz; spring constant: 18 N/m) was employed for electrical contact. At least 100 I-V curves were recorded under an applied out-of-plane magnetic field of 0.40 T.

## Pawley refinement

Unit cell parameters for chiral 2D c-MOFs were refined against PXRD data in the 2θ range of 2.5°–40° using the Reflex module in BIOVIA Materials Studio 2017 (version 17.1.0.48, Dassault Systèmes). Atom coordinates were fixed during the refinement process. The Pseudo-Voigt profile function was used for full-profile fitting. Final Rwp and Rp values were 2.70% and 1.80% for **(S)-Ph**, 4.01% and 2.33% for **(R)-Ph**, and 2.74% and 1.62% for **(S)-Na**, respectively.

## Data availability

The data supporting the findings of this study are available within the paper and its Supplementary Information files. Any additional datasets are available upon request from the corresponding author. Source data are provided with this paper.

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

## Acknowledgements

This work is financially supported by DFG projects (CRC-1415, no. 417590517; RTG 2861, No. 491865171), ERC Consolidator Grant (T2DCP, no. 819698), as well as the German Science Council and Center of Advancing Electronics Dresden (cfaed). S.F. gratefully acknowledges funding from the China Scholarship Council (202306240020). L.W. acknowledges the financial support from the European Research Council (ERC Starting Grant 'FastE-Chiral', 101162601). The authors acknowledge CFAED and Dresden Center for Nano-analysis (DCN) at TUD. These experiments were performed at BL11-NCD-SWEET beamline at ALBA Synchrotron with the collaboration of ALBA staff, and we would like to thank Dr. Cristian Huck for assistance in setting up the measurement.

## Author contributions

Y.L. and X.F. conceived the project. S.F. synthesized the ligands, pre-pared 2D c-MOFs, and conducted most of the structural, compositional, and property characterizations. C.W. fabricated the magnetic devices for CISS measurements. M.T. synthesized the ligand precursors. B.P. co-synthesized/characterized the chiral building blocks. X.H., L.S., and P.L. contributed to the discussion of synthesis and characterizations. F.A. performed the calculation and refinement of the chemical structure. R.H. conducted the electrical conductivity measurements. X.Z.W., X.W., L.W. and C.F. contributed to the CD measurement. D.W. conducted the HRTEM measurements. M.H., S.C.B.M. conducted the GIWAXS and data analysis. M.L. contributed to the SEM measurements. S.F., Y.L., and X.F. co-wrote the paper with the input of the other coauthors. All the authors discussed the results and commented on the manuscript.

## Funding

## Competing interests

The authors declare no competing interests.
