## [Transparent Peer Review file · Nature Communications]

Chiral two-dimensional conjugated metal-organic-frameworks with high spin polarization

Corresponding Author: Professor Yang Lu

Version 0:

Reviewer comments:

Reviewer #1

(Remarks to the Author)

[Note from editor: please see attached]

Reviewer #2

(Remarks to the Author)

The CISS effect, a focal point in spintronics research, demonstrates its unique role in regulating electron spin transmission. In this paper, the author reported the CISS effect in 2D c-MOFs, however, it is important to note that both 2D and 3D MOFs have already been explored within the context of CISS. The manuscript claims to report the highest spin current value to date; however, this claim lacks meaningful correlation with the efficiency of spin selectivity, which is a more relevant metric. For instance, the work by Rayo et al. (Nano Lett. 2020, 20, 12, 8476–8482) demonstrated 100% spin polarization across substantial crystal thicknesses (200–900 nm). Moreover, the characterization of the materials' CISS effect in the manuscript is quite limited, lacking device validation and a detailed mechanistic explanation. Overall, the routine nature of the study with many loopholes and lack of substantial insights/novelty does not contribute to progress the field forward and is not recommended for publication in a high-impact journal like "Nature Communications".

Reviewer #3

(Remarks to the Author)

This manuscript provides what appears to be fabrication and measurement of record or near-record breaking degrees of spin polarization through chiral 2D conjugated metal-organic-frameworks. Establishing a new record in observed spin polarized current transport through the utilization of these materials would represent a significant advance to a diverse array of applications including spin-tronics and chiral opto-electronic devices/materials platforms.

For the most part the work supports its conclusions and claims, although some additional reference information regarding the measurements and the filling out of all possible variations of the measurement is necessary to pull this manuscript into a complete work for publication in a high impact journal such as Nature Communications.

I see no glaring flaws in the data analysis and conclusions. It was, however, unclear to me why measurement of CISS via magneto-conductive AFM on (R)-NaEt materials were not included in the manuscript. It would seem relevant to include all possible variations of R and S PhEt and NaEt. The authors include both R and S handedness for PhEt, but not for NaEt. For completeness, I would suggest adding this characterization to tell the full story of these materials. Further, while the authors show that they can control the thickness of their MOF thin films, they do not provide a characterization of the thickness dependence of CISS transport through their thin films. I would suggest adding this analysis if possible. At the very least, the thickness of all the films measured should be more precisely noted. Stating approximately ~100 nm is too vague. For example, the authors observe a slightly higher spin polarization for NaEt, but this could potentially be due to a thickness variation of the sample rather than an intrinsic property of NaEt. Finally, for the mc-AFM measurements the authors should clarify how the measurements were performed. Do the I-V curves represent measurements from a number of spatially distinct locations on the surface- or simply measurements performed a number of times at the same location?

Minor Edits/Suggestions:

Main text Figure 4, panel a: The chirality of the spirals doesn't change between R and S handedness. Check the chirality of the cartoon image and change as necessary so that it actually varies with the correct chirality

Supporting Materials Line 293: Add colorbar for thickness of measurements. Adding line traces showing the thickness and roughness of the sample would be even more helpful.

Version 1:

Reviewer comments:

Reviewer #1

(Remarks to the Author)

The authors have thoroughly addressed all my questions and concerns. I recommend its acceptance for publication in its current form.

Reviewer #3

(Remarks to the Author)

Having read through the authors responses to my points of criticism, I feel that they have adequately addressed my points of concern and would support publication in Nature Communications. While I understand some of the criticisms noted by the other reviewers as to the inclusion of a broader model of the spin dependent transport, I would argue that our study and understanding of CISS transport effects is still in nascent stages. A demonstration combining a unique MOF system with an observation of a high degree of spin polarized transport will represent a high impact work, even without a bespoke theoretical framework describing the fundamental mechanisms of this transport. Indeed, the development of complete theoretical models of the CISS effect has been ongoing for the past decade, so a focus on the experimental observations made by this group provides a key system and data set for theorists.

One very minor request for revision- especially if this is going to be published in a high impact journal with many scientists reviewing the data. For all of the AFM images, please use a filter to remove the image tilt. It appears as if you are either showing raw data, or incorrectly applying a linear fit to a fit of your images step edge which makes the AFM image look tilted. You should mask off the portions of your images that have a differing height and then perform a plane fit so the relevant areas that you are interested in showing appear flat and not tilted. Any commercial AFM image processing software will have this available. It will only take a few min to do, but will improve the visual communication of the data that you are looking for in your AFM figures. While the images still convey the information they are supposed to, they are harder to visually interpret and may confuse some readers.

Response for Reviewer #1:

General Comment: *This work presents a novel side chain-induced chirality amplification (SICA) strategy to synthesize 2D chiral conjugated MOFs (c-MOFs) with high spin polarization. By integrating chiral side chains into the hexaiminotriindole (HATI) ligand backbone, the authors successfully transferred molecular chirality to the extended framework while maintaining structural symmetry. The chirality degree was tunable via steric hindrance, directly enhancing bulk spin polarization. The materials were rigorously characterized, including magnetic-conductive AFM (mc-AFM), confirming exceptional conductivity and CISS effect. Overall, this work demonstrates certain novelty and merits publication after major revisions.*

Our response:

We wish to express our sincere gratitude to Reviewer #1 for the positive evaluation and for highlighting the key advancements contained in our work. We took full advantage of all suggestions, which turned out to be instrumental in improving our manuscript.

1. *Could the authors provide more detailed guidelines on selecting side-chain groups (beyond PhEt/NaEt) to optimize steric effects? A systematic study comparing alkyl, aryl, and bulkier substituents (e.g., *t*-butyl, anthracene) might reveal whether distortion angles correlate linearly with spin polarization or if an optimal threshold exists for maximizing chirality transfer without compromising conductivity.*

Our response:

We appreciate the valuable comments. In our previous work, we investigated alkyl chains with varying degrees of steric hindrance, including methyl, propyl, isopropyl, allyl, and butyl substituents (*Nat. Commun.* **2022**, *13*, 7240; *J. Am. Chem. Soc.* **2024**, *146*, 2574-2582). Although bulky groups such as isopropyl have been introduced, the resulting 2D c-MOFs remained racemic, as these side chains are achiral and do not impart significant chiral characteristics. In parallel, we introduced a chiral *sec*-butyl group and successfully synthesized the corresponding ligand. Unfortunately, despite numerous attempts, we were unable to obtain highly crystalline MOFs (Figure R1). We attribute this to the long-branched alkyl chain of the *sec*-butyl group, which significantly increases the solubility of the ligand and thereby hinders the crystallization process required for MOF formation. Further, the introduction of chiral side groups with greater steric hindrance (e.g., anthrylethyl) is challenging to achieve due to limitations in synthetic methods. Our synthetic strategy relies on a one-step Mitsunobu reaction to introduce the side chain on the conjugated backbone. This approach requires a chiral pure α -methyl-2-anthracenemethanol precursor, which is not commercially available, thereby posing a significant challenge for introducing chiral anthracene derivatives with greater steric hindrance. Nevertheless, based on the promising results from our (S)-PhEt, (R)-PhEt, and (S)-NaEt systems, we believe that introducing bulkier chiral side chains—provided that the synthesis of highly crystalline MOFs is not compromised—could further amplify the chiral properties of the resulting MOFs.

Figure R1. PXRD pattern of the MOF prepared from a conjugated ligand bearing chiral *sec*-butyl substituents. The MOF synthesis was performed using 1.5 equivalents of Ni(OAc)₂ and 150 equivalents of NaOAc as the metal precursor and base, respectively.

2. How does the chiral distortion propagate from the side chain to the MOF framework at the atomic level? DFT calculations or single-crystal XRD analysis of intermediate HATI derivatives could clarify whether the propeller-like twist is rigid or dynamic, and whether cooperative effects (e.g., H-bonding, π -stacking) between adjacent ligands enhance chirality amplification.

Our response:

Indeed, we have performed relevant DFT calculations to investigate the structural features of the chiral 2D c-MOFs. The results indicate that these frameworks adopt an AA-eclipsed stacking geometry. The introduction of chiral groups with different configurations induces a fan-like distortion of the ligand backbone, with the rotational direction corresponding to the handedness of the chiral group. Notably, bulkier chiral substituents lead to a greater degree of distortion. For example, replacing the chiral PhEt group with a more sterically hindered NaEt group increases the distortion angle from 2.3° to 7.0°, thereby producing a larger fan-blade structure. The coordination polymerization of chiral HATI ligands with metal ions results in the formation of chiral 2D c-MOFs, wherein the chirality of individual ligands is transferred and amplified throughout the framework. As illustrated in the side-view analysis (Figure R2), in addition to the π - π interactions between triindole backbones, the aromatic rings of side chains from adjacent layers participate in interlayer π - π stacking. This interaction further locks the propeller-like configuration and makes the overall chiral framework very rigid. To clarify this point, we have added the corresponding discussion to the revised manuscript.

Figure R2. Side-view of the simulated structures of the 2D c-MOFs: (a) (S)-Ph, (b) (R)-Ph, (c) (S)-Na.

ACTION:

Page 7, Line 151-153

“Specifically, the aromatic rings on the side chains contribute to interlayer π - π stacking, which stabilizes the twisted conformation and helps maintain both the chiral framework and its rigidity.”

3. Have the authors evaluated the stability of these chiral 2D c-MOFs under ambient/moisture conditions or thermal stress? Given the potential for π -conjugated backbones to undergo oxidation or hydrolysis, stability data would be critical for assessing practical applicability in spintronic devices.

Our response:

We would like to thank the reviewer for raising this important concern. To evaluate the stability of the chiral 2D c-MOFs, we conducted long-term storage tests under ambient (~40% RH) and moisture-rich (~95% RH) environments for 30 days. After storage, PXRD measurements were performed on both the freshly prepared samples and the stored ones. The PXRD patterns of all three samples remained unchanged and showed excellent overlap with those of the fresh samples, indicating that the crystallinity and structural integrity of the materials were preserved. Furthermore, the chiral 2D c-MOFs were synthesized at 85 °C in a mixed solvent system of water and DMSO, yielding highly crystalline and stable products, which reflects their inherent thermal robustness. These findings collectively demonstrate the excellent thermal and environmental stability of this

class of chiral 2D c-MOFs. The corresponding discussion and supporting data have been incorporated into the revised manuscript and Supplementary Information.

Figure R3. PXRD patterns of freshly prepared (S)-Ph 2D c-MOF and samples stored under ambient (~40% RH) and moisture-rich (~95% RH) environments for 30 days.

ACTION:

Page 10, Line 200-202

SI, Page 18

“Notably, these chiral 2D c-MOFs exhibit excellent ambient and moisture stability, retaining high crystallinity after 30 days of storage at room temperature (Supplementary Fig. S10).”

Supplementary Figure 10. Stability of chiral 2D c-MOFs. PXRD patterns of freshly prepared (S)-Ph 2D c-MOF and samples stored under ambient (~40% RH) and moisture-rich (~95% RH) environments for 30 days.

4. While *mc*-AFM confirms high spin polarization, how do these MOFs perform in real spintronic architectures (e.g., spin valves, MTJs)? Thin-film fabrication and electrode compatibility (e.g.,

interfacial spin scattering) could pose challenges, and have preliminary device tests been conducted? A comparison with known chiral polymers/oxides would contextualize the MOFs' advantages.

Our response:

We would like to thank the reviewer for this insightful comment. Compared with conventional chiral polymers or oxides, one of the major challenges associated with chiral 2D c-MOFs is their significant surface roughness produced by the solution synthesis approach, which arises from their inherent crystallinity when processed into thin films (*Nat. Mater.* **2023**, *22*, 322-328; *Angew. Chem. Int. Ed.* **2024**, *63*, e202412283; *Angew. Chem. Int. Ed.* **2020**, *59*, 14671-14676; *Adv. Mater.* **2015**, *27*, 1924-1927; *ACS Nano* **2022**, *16*, 12145-12155). To address this challenge, we made considerable efforts in this work. By precisely controlling oxygen diffusion, we successfully synthesized chiral 2D c-MOF films using air-liquid interfacial synthesis. This method significantly reduces the surface roughness compared to traditional techniques. For instance, films prepared via the commonly used layer-by-layer method typically exhibit a surface roughness up to 150 nm (*Electrochim. Acta*, **2023**, *469*, 143152), whereas our synthesized films show a reduced surface roughness ranging from 7 to 62 nm (Figure R4). Despite this improvement, the surface roughness of our chiral 2D c-MOF films is still notably higher than that of reported chiral polymers and oxides, which generally exhibit roughness values below 10 nm. This relatively high roughness poses a significant limitation, as it hampers the formation of well-defined interfaces that are crucial for spintronic device applications.

In future work, we will continue to explore methods for further reducing surface roughness. Our recent studies have demonstrated that chemical vapor deposition (CVD) can produce smoother 2D c-MOF films (*Nat. Synth.* **2024**, *3*, 715-726), making it a promising approach for the fabrication of spintronic devices. Achieving 2D c-MOF films with high-quality, high-crystallinity, and low-roughness is essential for advancing their application in spintronics, and this will remain a central focus of our ongoing research. The corresponding discussion has been added to the revised manuscript.

Figure R4. Surface roughness of (S)-Ph film varying thickness of 31.5, 101.3 nm to 308.5 nm. The dashed square in the inset indicates the area used to calculate the Rq roughness.

ACTION:

Page 14, Line 302-304

“The synthesis of high-crystallinity chiral 2D c-MOF films, along with the fabrication of high-performance spintronic devices, will remain the focus of our future efforts.”

Response for Reviewer #2:

General Comment: *The CISS effect, a focal point in spintronics research, demonstrates its unique role in regulating electron spin transmission. In this paper, the author reported the CISS effect in 2D c-MOFs, however, it is important to note that both 2D and 3D MOFs have already been explored within the context of CISS. The manuscript claims to report the highest spin current value to date; however, this claim lacks meaningful correlation with the efficiency of spin selectivity, which is a more relevant metric. For instance, the work by Rayo et al. (Nano Lett. 2020, 20, 12, 8476–8482) demonstrated 100% spin polarization across substantial crystal thicknesses (200–900 nm). Moreover, the characterization of the materials' CISS effect in the manuscript is quite limited, lacking device validation and a detailed mechanistic explanation. Overall, the routine nature of the study with many loopholes and lack of substantial insights/novelty does not contribute to progress the field forward and is not recommended for publication in a high-impact journal like “Nature Communications”.*

Our response:

We appreciate Reviewer #2's comments on the pivotal aspects of our study. While acknowledging the concerns raised regarding novelty and characterization depth, we wish to highlight that our work fundamentally differs from prior MOF-based CISS systems through its novel conductive chiral 2D crystals platform, which drives unprecedented spin-current enhancement. The subsequent responses will systematically address: (1) Comparison with previous 2D and 3D MOF systems; (2) Interpretation of spin current versus spin selectivity; and (3) Characterization and deep understanding of chiral 2D c-MOFs' CISS effect. We contend that these elements collectively substantiate the significant advance offered by our chiral 2D c-MOFs.

1) Comparison with previous 2D and 3D MOF systems.

Compared with previously reported 2D and 3D MOF systems, our work introduces a molecular-level design strategy to impart chirality into organic 2D crystals and achieve chirality amplification through side-chain engineering. Chiral 2D crystals are of great importance not only for chiral-induced spin selectivity (CISS), but also for applications in optoelectronics and spintronics (*Science*, **1998**, 282, 913-915; *Chem. Soc. Rev.* **2020**, 49, 6248-6272; *Proc. Natl. Acad. Sci. U.S.A.* **2020**, 117, 10721-10726; *Nat. Rev. Phys.* **2021**, 3, 328-343; *Nat. Rev. Chem.* **2019**, 3, 250-260). However, due to the intrinsic requirement of high symmetry in the design of extended 2D frameworks, the molecular-level introduction of chirality into such systems remains a formidable challenge (*Chem* **2022**, 8, 1822-1854). In this work, our primary focus is to develop a generalizable strategy for constructing new chiral 2D crystals. To this end, we propose a rational molecular design approach to introduce chirality into 2D frameworks, and further amplify this chirality through side-chain engineering. The

chiral 2D c-MOFs developed in this study exhibit essential differences from conventional 2D and 3D MOFs. This strategy is not only applicable to 2D c-MOFs, but also holds great potential for a range of carbon-rich organic 2D crystals, including 2D conjugated polymers, 2D covalent organic frameworks (COFs), and 2D hydrogen-bonded organic frameworks (HOFs). Ultimately, we successfully synthesized the first example of chiral 2D c-MOF with well-defined chiral architectures.

2) Interpretation of spin current versus spin selectivity.

For the CISS effect, it is important to emphasize that enhancing the electrical conductivity of chiral materials, which can contribute to the generation of spin-polarized current, is crucial for improving spin selectivity, as demonstrated in recent studies (*J. Am. Chem. Soc.* **2023**, *145*, 26791-26798). Furthermore, the distinctive d- π conjugated networks in our 2D c-MOFs give rise to strong spin-orbit coupling, which plays a vital role in enhancing the intrinsic spin polarization. (*Nat. Mater.* **2021**, *20*, 222-228; *Chem. Soc. Rev.* **2021**, *50*, 2764-2793; *Angew. Chem. Int. Ed.* **2016**, *55*, 3566-3579). As a proof of concept, our chiral 2D c-MOFs demonstrate high electrical conductivity and exhibit substantial spin-polarized current, offering clear advantages in CISS-related applications. Conventional 2D or 3D MOFs typically exhibit low bulk electrical conductivity (*Nat. Commun.* **2018**, *9*, 2637; *Chem. Soc. Rev.* **2017**, *46*, 3185-3241; *Angew. Chem. Int. Ed.* **2016**, *55*, 3566-3579). For instance, a 3D MOF reported by Rayo et al. (*Nano Lett.* **2020**, *20*, 8476-8482) showed a spin current of only 0.85 nA at 1.5 V. In comparison, our chiral 2D c-MOFs exhibited a significantly higher spin current of 241.9 nA at the same voltage, which is approximately 285 times greater than that of the 3D MOF. This comparison clearly demonstrates the substantial improvement in spin current performance achieved by our chiral 2D c-MOFs over conventional 3D MOFs.

3) Characterization and deep understanding of chiral 2D c-MOFs' CISS effect.

To gain a deeper understanding of the CISS effect in chiral 2D c-MOFs, we conducted comprehensive characterizations and mechanistic investigations. During the mc-AFM measurements, more than 100 independent measurements were performed for each sample batch. These measurements yielded stable and reproducible current values, in contrast to many previous studies that report only a single representative curve, raising concerns about data reliability (Figure R5). Our results highlight that the high electrical conductivity of chiral 2D c-MOFs plays a critical role in achieving reliable and reproducible spin polarization measurements. In addition, we further performed thickness-dependent CISS measurements by varying the film thickness from 30 to 300 nm. As shown in Figure R6, the measured spin polarization values of (S)-Ph films at thicknesses of 31.5 nm, 101.3 nm, and 308.4 nm are 86.8% \pm 0.2%, 91.6% \pm 0.1%, and 95.6% \pm 0.1%, respectively (Figure R6). The observed spin polarization shows a clear dependence on film thickness, which is consistent with previous reports and correlates with the length of the chiral transport pathways (*Science*, **2011**, *331*, 894-

897; *Sci. Adv.* **2022**, *8*, eabq2727; *J. Phys. Chem. C* **2020**, *124*, 19, 10776-10782; *J. Phys. Chem. C* **2015**, *119*, 26, 14542-14547). This thickness-dependent behavior is attributed to the extended chiral pathways and the intrinsic face-on stacking of the MOF layers, which together facilitate out-of-plane charge and spin transport (*Phys. Rev. B* **2013**, *88*, 054417; *Phys. Rev. Lett.* **2021**, *127*, 126602). Furthermore, the high crystallinity of the films enables long-range charge transfer, contributing to the high spin polarization observed. Finally, while we are actively investigating additional aspects of spin transport in chiral materials, including the anisotropic spin transport properties of chiral c-MOFs, we consider a detailed discussion of these topics to be beyond the scope of this study.

In conclusion, the novelty and significance of our work are further supported by the comments from Reviewer #1 and Reviewer #3. Reviewer #1 acknowledged the originality of our approach, stating: “By integrating chiral side chains into the hexaiminotriindole (HATI) ligand backbone, **the authors successfully transferred molecular chirality to the extended framework while maintaining structural symmetry.** The chirality degree was tunable via steric hindrance, directly enhancing bulk spin polarization.” This highlights the conceptual innovation and fundamental importance of our molecular design strategy.

In addition, the exceptional conductivity of our materials and the scarcity of chiral 2D c-MOFs are emphasized by Reviewer #3, who commented: “This manuscript provides what appears to be fabrication and measurement of **record or near-record breaking degrees of spin polarization through chiral 2D conjugated metal-organic-frameworks.** Establishing a new record in observed spin-polarized current transport through the utilization of these materials would represent a significant advance to a diverse array of applications including spintronics and chiral optoelectronic devices/materials platforms.”

Taken together, these insights affirm the novelty, relevance, and potential impact of our work. We therefore believe that our molecular engineering strategy offers a promising new avenue for the development of chiral 2D crystals for spintronics and beyond. Moving forward, we will continue to explore the synthesis of high-crystallinity chiral 2D c-MOF films aimed at the fabrication of high-performance spintronic devices.

Figure R5. Measurement positions during mc-AFM testing. Green crosses denote individual locations on the sample surface where I-V spectra were collected.

Figure R6. Thickness-dependent CISS measurements. I-V curves of (S)-Ph films with varying thickness.

Response for Reviewer #3:

General Comment: *This manuscript provides what appears to be fabrication and measurement of record or near-record breaking degrees of spin polarization through chiral 2D conjugated metal-organic-frameworks. Establishing a new record in observed spin polarized current transport through the utilization of these materials would represent a significant advance to a diverse array of applications including spin-tronics and chiral opto-electronic devices/materials platforms. For the most part the work supports its conclusions and claims, although some additional reference information regarding the measurements and the filling out of all possible variations of the measurement is necessary to pull this manuscript into a complete work for publication in a high impact journal such as Nature Communications.*

Our response:

We thank Reviewer #3 for the encouraging assessment and for recognizing the significance of our work in advancing the development of chiral 2D crystals for spintronics and related technologies. We greatly appreciate the reviewer's insightful suggestions, which have been invaluable in

improving the clarity and completeness of our manuscript. We have carefully addressed the comments and incorporated the necessary additions to strengthen the experimental support for our conclusions.

1. I see no glaring flaws in the data analysis and conclusions. It was, however, unclear to me why measurement of CISS via magneto-conductive AFM on (R)-NaEt materials were not included in the manuscript. It would seem relevant to include all possible variations of R and S PhEt and NaEt. The authors include both R and S handedness for PhEt, but not for NaEt. For completeness, I would suggest adding this characterization to tell the full story of these materials.

Our response:

At the outset, we also intended to synthesize the (R)-NaEt enantiomer to complete the full set of chiral analogues. However, this approach was hindered by the high cost of enantiomerically pure (R)-1-(2-naphthyl)ethanol, which exceeds €1000 for 5 grams. Since this chiral alcohol is required in the first step of the synthetic route, the cost poses a significant barrier to preparing the practical-scale preparation of the corresponding monomer.

Furthermore, the successful synthesis and characterization of (S)-PhEt and (R)-PhEt allowed us to demonstrate the enantioselective transfer of molecular chirality from the side chains to the ligand backbone and ultimately to the extended 2D framework, all while maintaining overall structural symmetry. In addition, the comparison between (S)-PhEt and (S)-NaEt confirmed that the degree of chirality can be amplified through side-chain steric enhancements, resulting in enhanced spin polarization. Although we were unable to obtain the (R)-NaEt derivative due to these synthetic limitations, our study still clearly supports the feasibility of using rational side-chain engineering to access chiral 2D crystals with tunable handedness and amplified chirality at the molecular level.

2. Further, while the authors show that they can control the thickness of their MOF thin films, they do not provide a characterization of the thickness dependence of CISS transport through their thin films. I would suggest adding this analysis if possible.

Our response:

We appreciate the reviewer for this insightful comment. Following your suggestion, we performed thickness-dependent CISS measurements by varying the film thickness from 30 to 300 nm. As shown in Figure R7, the measured spin polarization values of (S)-Ph at thicknesses of 31.5 nm, 101.3 nm, and 308.4 nm are $86.8\% \pm 0.2\%$, $91.6\% \pm 0.1\%$, and $95.6\% \pm 0.1\%$, respectively. The spin polarization in the chiral MOF films exhibits a clear dependence on thickness, which correlates with the length of the chiral structure. These observations, including the sign of spin polarization and its dependence on molecular length, are consistent with previously reported studies (*Science*, **2011**, 331, 894-897; *Sci. Adv.* **2022**, 8, eabq2727; *J. Phys. Chem. C* **2020**, 124, 19, 10776-10782; *J. Phys. Chem. C* **2015**, 119, 26, 14542-14547). This strong thickness dependence is likely due to the extended length of the chiral structures within the MOF films. The intrinsic face-on stacking arrangement facilitates charge carriers to move along the out-of-plane direction, promoting efficient

transport of spin-polarized current through the continuous chiral pathways and ensuring effective interaction with the chiral environment (*Phys. Rev. B* **2013**, *88*, 054417; *Phys. Rev. Lett.* **2021**, *127*, 126602). Furthermore, the high crystallinity of the chiral MOF films supports long-range charge transport, which contributes to the ultrahigh spin polarization values observed. The corresponding results and analysis have been added to the revised manuscript and Supplementary Information.

Figure R7. Thickness-dependent CISS measurements. I-V curves of (S)-Ph films with varying thickness.

ACTION:

Page 13, Line 270-284

SI, Page 20 and 24

“Moreover, we conducted thickness-dependent CISS measurements using films with thicknesses of 31.5 nm, 101.3 nm, and 308.5 nm (Supplementary Fig. S14). As shown in Supplementary Fig. S21, the measured spin polarization values of (S)-Ph at thicknesses of 31.5 nm, 101.3 nm and 308.4 nm are 86.8% ± 0.2%, 91.6% ± 0.1%, and 95.6% ± 0.1%, respectively. The spin polarization in the chiral MOF films exhibits a clear dependence on thickness, which correlates with the length of chiral structure. These results, including both the sign of the polarization and its dependence on length, are in good agreement with previously reported studies.^{8, 10, 47, 48} This strong thickness dependence is likely due to the extended length of the chiral structures within the MOF films. The inherent face-on stacking arrangement enables charge carriers to move along the out-of-plane direction, promoting efficient transport of spin-polarized current through the continuous chiral pathways and ensuring effective interaction with the chiral environment.^{49, 50} In addition, the high crystallinity of the chiral MOF films facilitates long-range charge transport, which contributes to the observed ultrahigh spin polarization values.”

Supplementary Figure 14. AFM images of chiral MOF films. By modulating the air diffusion rate, the film thickness was precisely controlled from 31.5 nm to 308.5 nm.

Supplementary Figure 21. Thickness-dependent CISS measurements. The measured spin polarization values of (S)-Ph at thicknesses of 31.5 nm, 101.3 nm and 308.4 nm are $86.8\% \pm 0.2\%$, $91.6\% \pm 0.1\%$, and $95.6\% \pm 0.1\%$, respectively. These results clearly demonstrate the strong dependence of spin polarization on film thickness in chiral MOF films. Due to synthesis limitations, it was challenging to synthesize chiral MOF films with thicknesses exceeding 400 nm.

3. At the very least, the thickness of all the films measured should be more precisely noted. Stating approximately ~ 100 nm is too vague. For example, the authors observe a slightly higher spin polarization for NaEt, but this could potentially be due to a thickness variation of the sample rather than an intrinsic property of NaEt.

Our response:

We appreciate the reviewer for this valuable suggestion. In response to your comment, we have carefully revised all thickness-related descriptions throughout the manuscript and Supplementary Information to improve accuracy and consistency. In addition, we conducted the CISS measurements using chiral 2D c-MOF films with very comparable thicknesses: 101.3 nm for (S)-Ph, 100.8 nm for (R)-Ph, and 101.7 nm for (S)-Na (Figure R8). This controlled thickness allows us to attribute the observed amplification of spin polarization primarily to the intrinsic chiral properties of (S)-Na. The corresponding thickness data have been added to the revised manuscript and Supplementary Information.

Figure R8. Thickness of chiral 2D c-MOF films: 101.3 nm for (S)-Ph, 100.8 nm for (R)-Ph, and 101.7 nm for (S)-Na.

ACTION:

Page 11, Line 223-226

SI, Page 20 and 23

“To address this issue as a proof-of-concept, chiral 2D c-MOF films were synthesized with thicknesses of 101.3, 100.8, and 101.7 nm for (S)-Ph, (R)-Ph, and (S)-Na, respectively, using Ni substrates for subsequent CISS measurements (see Supplementary Information, Fig. S12-S19 for synthesis details).”

Supplementary Figure 14. AFM images of chiral MOF films. By modulating the air diffusion rate, the film thickness was precisely controlled from 31.5 nm to 308.5 nm.

Supplementary Figure 19. Thickness of chiral 2D c-MOFs. Chiral 2D c-MOF films were synthesized with thicknesses of 101.3, 100.8, and 101.7 nm for (S)-Ph, (R)-Ph, and (S)-Na, respectively.

4. Finally, for the mc-AFM measurements the authors should clarify how the measurements were performed. do the I-V curves represent measurements from a number of spatially distinct locations on the surface- or simply measurements performed a number of times at the same location?

Our response:

We thank the reviewer for this valuable comment regarding the spatial distribution of the mc-AFM measurements. Owing to the high homogeneity of the synthesized chiral 2D c-MOF films, measurement points were randomly selected across the sample surface to ensure spatial consistency during I-V measurements. Figure R9 presents an AFM image of the measurement locations, where each green cross denotes a selected point. For each sample batch, over 100 independent measurements were performed, consistently yielding stable and reproducible current values. Additionally, in response to the reviewer's comment, the relevant measurement details have been added to the revised Supplementary Information.

Figure R9. Measurement positions during mc-AFM testing. Green crosses denote individual

locations on the sample surface where I-V spectra were collected.

ACTION:

SI, Page 5-6

Magnetic-conducting Atomic Force Microscope (mc-AFM)

Sample Preparation

A silicon wafer with a 500 nm thermal oxide layer was used as the substrate. A 5 nm titanium adhesion layer and an 80 nm nickel layer were sequentially deposited, followed by a 5 nm gold protective layer. The resulting Ni/Au substrate facilitates spin polarization of electrons injected into the chiral MOF films under an external magnetic field. Prior to film transfer, the substrate was sequentially cleaned by immersion in ethanol and acetone for 5 minutes each, then dried under a nitrogen stream. Chiral 2D c-MOF films were subsequently transferred onto the prepared Ni/Au substrate for mc-AFM measurements.

Measurement of the CISS Effect

Current-voltage (I-V) characteristics were measured using a conductive atomic force microscopy (c-AFM) mode on a Park NX10 system (Park Systems). Measurements were performed in pin-point mode, in which a voltage bias is applied through the conductive tip while minimizing lateral forces, thereby enhancing spatial resolution and reducing sample deformation or displacement. A platinum-coated conductive probe (model: 25Pt300B; resonance frequency: 20 kHz; spring constant: 18 N/m) was employed for electrical contact. At least 100 I-V curves were recorded under an applied out-of-plane magnetic field of 0.40 T.

5. Minor Edits/Suggestions: Main text Figure 4, panel a: The chirality of the spirals doesn't change between R and S handedness. Check the chirality of the cartoon image and change as necessary so that it actually varies with the correct chirality. Supporting Materials Line 293: Add colorbar for thickness of measurements. Adding line traces showing the thickness and roughness of the sample would be even more helpful.

Our response:

We are grateful to Reviewer #3 for the thoughtful suggestions that have helped us improve the clarity of our manuscript. To better distinguish the chirality between the R- and S-enantiomers, we have revised the schematic illustration in Figure 4 to show distinct spiral orientations (Figure R10). Additionally, in response to the reviewer's suggestion, we have revised color bar indicating film thickness, along with line trace data and roughness to the revised Supplementary Information (Figure R11).

Figure R10. Schematic illustration of the tunneling current in a parallel configuration for (S)-Ph, (R)-Ph, and (S)-Na.

Figure R11. AFM images of chiral MOF films. By modulating the air diffusion rate, the film thickness was precisely controlled from 31.5 nm to 308.5 nm.

Response for Reviewer #1:

General Comment: *The authors have thoroughly addressed all my questions and concerns. I recommend its acceptance for publication in its current form.*

Our response:

Thank you very much for your recognition of our work. We are deeply grateful for your constructive comments and suggestions, which have greatly helped us to further polish and refine the manuscript. Your feedback has enabled us to present our study in a clearer and more comprehensive manner, and we are very pleased that you find the revised version suitable for publication in its current form. We sincerely appreciate your support and encouragement, which motivates us to continue our research and share our findings with the scientific community.

Response for Reviewer #3:

General Comment: *Having read through the authors responses to my points of criticism, I feel that they have adequately addressed my points of concern and would support publication in Nature Communications. While I understand some of the criticisms noted by the other reviewers as to the inclusion of a broader model of the spin dependent transport, I would argue that our study and understanding of CISS transport effects is still in nascent stages. A demonstration combining a unique MOF system with an observation of a high degree of spin polarized transport will represent a high impact work, even without a bespoke theoretical framework describing the fundamental mechanisms of this transport. Indeed, the development of complete theoretical models of the CISS effect has been ongoing for the past decade, so a focus on the experimental observations made by this group provides a key system and data set for theorists.*

Our response:

We sincerely thank you for your recognition of our work. As you have rightly pointed out, further efforts are indeed needed to gain deeper insights into the transport mechanisms underlying the CISS effect. Providing a broader library of experimental systems, particularly those based on 2D conjugated MOFs and other related crystalline chiral materials, has always been one of our long-term goals. We will continue to devote ourselves to advancing the study of CISS and aim to make meaningful contributions to the development of this field.

We are also very grateful for your valuable comments, which have helped us polish the manuscript and enhance its professionalism. Your thoughtful support and encouragement strongly motivate us to pursue further progress and to share high-quality results with the scientific community.

1. One very minor request for revision- especially if this is going to be published in a high impact journal with many scientists reviewing the data. For all of the AFM images, please use a filter to remove the image tilt. It appears as if you are either showing raw data, or incorrectly applying a linear fit to a fit of your images step edge which makes the AFM image look tilted. You should mask

off the portions of your images that have a differing height and then perform a plane fit so the relevant areas that you are interested in showing appear flat and not tilted. Any commercial AFM image processing software will have this available. It will only take a few min to do, but will improve the visual communication of the data that you are looking for in your AFM figures. While the images still convey the information they are supposed to, they are harder to visually interpret and may confuse some readers.

Our response:

We sincerely thank you for your professional advice regarding the AFM images. Following your suggestion, we have carefully reprocessed all relevant AFM figures in both the main manuscript and the Supplementary Information by applying appropriate flattening and plane-fitting procedures, so that the images more clearly convey the intended information. We will also make sure to pay closer attention to the professional standards in applying different characterization techniques in our future work. We truly appreciate your constructive comment, which has helped us to improve the quality and clarity of our manuscript.

Figure R1. AFM images of chiral MOF films. By modulating the air diffusion rate, the film thickness was precisely controlled from 31.5 nm to 308.5 nm.

Figure R2. Thickness of chiral 2D c-MOFs. Chiral 2D c-MOF films were synthesized with thicknesses of 101.3, 100.8, and 101.7 nm for (S)-Ph, (R)-Ph, and (S)-Na, respectively.

This work presents a novel side chain-induced chirality amplification (SICA) strategy to synthesize 2D chiral conjugated MOFs (c-MOFs) with high spin polarization. By integrating chiral side chains into the hexaiminotriindole (HATI) ligand backbone, the authors successfully transferred molecular chirality to the extended framework while maintaining structural symmetry. The chirality degree was tunable via steric hindrance, directly enhancing bulk spin polarization. The materials were rigorously characterized, including magnetic-conductive AFM (mc-AFM), confirming exceptional conductivity and CISS effect. Overall, Overall, this work demonstrates certain novelty and merits publication after major revisions.

1. Could the authors provide more detailed guidelines on selecting side-chain groups (beyond PhEt/NaEt) to optimize steric effects? A systematic study comparing alkyl, aryl, and bulkier substituents (e.g., t-butyl, anthracene) might reveal whether distortion angles correlate linearly with spin polarization or if an optimal threshold exists for maximizing chirality transfer without compromising conductivity.
2. How does the chiral distortion propagate from the side chain to the MOF framework at the atomic level? DFT calculations or single-crystal XRD analysis of intermediate HATI derivatives could clarify whether the propeller-like twist is rigid or dynamic, and whether cooperative effects (e.g., H-bonding, π -stacking) between adjacent ligands enhance chirality amplification.
3. Have the authors evaluated the stability of these chiral 2D c-MOFs under ambient/moisture conditions or thermal stress? Given the potential for π -conjugated backbones to undergo oxidation or hydrolysis, stability data would be critical for assessing practical applicability in spintronic devices.
4. While mc-AFM confirms high spin polarization, how do these MOFs perform in real spintronic architectures (e.g., spin valves, MTJs)? Thin-film fabrication and electrode compatibility (e.g., interfacial spin scattering) could pose challenges, and have preliminary device tests been conducted? A comparison with known chiral polymers/oxides would contextualize the MOFs' advantages.